# Genetic Polymorphism and Phylogenetics of *Aedes aegypti* from Sudan Based on *ND4* Mitochondrial Gene Variations

**DOI:** 10.3390/insects13121144

**Published:** 2022-12-12

**Authors:** Sara Abdelrahman Abuelmaali, Jamsari Amirul Firdaus Jamaluddin, Mushal Allam, Hind Mohamed Abushama, Dia Eldin Elnaiem, Kheder Noaman, Silas Wintuma Avicor, Intan Haslina Ishak, Mustafa Fadzil Farid Wajidi, Zairi Jaal, Nur Faeza Abu Kassim

**Affiliations:** 1129 Medical Entomology Laboratory, School of Biological Sciences, Universiti Sains Malaysia, Penang 11800, Malaysia; 2National Public Health Laboratory, Federal Ministry of Health, Khartoum 11115, Sudan; 3College of Medicine and Health Sciences, United Arab Emirates University, P.O. Box 15551 Abu Dhabi, United Arab Emirates; 4Department of Zoology, Faculty of Science, University of Khartoum, Khartoum 321, Sudan; 5Department of Natural Sciences, University of Maryland Eastern Shore, Maryland, MD 21853, USA; 6National Center for Research, Tropical Medicine Research Institute, Khartoum 1304, Sudan; 7Entomology Division, Cocoa Research Institute of Ghana, New Tafo-Akim P.O. Box 8, Ghana; 8Molecular Entomology Research Group, Universiti Sains Malaysia, Penang 11800, Malaysia; 9Vector Control Research Unit, School of Biological Sciences, Universiti Sains Malaysia, Penang 11800, Malaysia

**Keywords:** mitochondrial DNA, NADH dehydrogenase subunit 4 (*ND4*), *Aedes aegypti* subspecies, genetic variation, genetic structure, Sudan

## Abstract

**Simple Summary:**

Arthropod-borne viruses (arboviruses) place a heavy social and economic burden on society by infecting millions of people and causing widespread animal and human sickness. These viruses are often transmitted to their vertebrate hosts by arthropod vectors. The mosquito *Aedes aegypti* is thought to be the principal vector for arboviral diseases such as dengue, yellow fever, zika, chikungunya, and others. *Aedes aegypti*, which is believed to have its origins in Africa, is known to have two subspecies or forms that are differentiated from one another in terms of behavior, transmitting power, and dispersal. The study of genetic structure and variations among different forms/subspecies of disease vectors sheds important light on their biology, behavior, genetic mixing, and capacity to spread disease. To better understand the genetic polymorphism of the two forms of *Aedes aegypti*, this study looked at various populations of the mosquito in different locations in Sudan. The findings revealed that the genetic variance between the two subspecies was insignificant, and they shared some components in the gene/marker we used in the study. This information will be required to improve the control strategies against the *Aedes aegypti* vector in the area.

**Abstract:**

This study investigated the genetic differences between *Aedes aegypti* subspecies (*Aedes aegypti aegypti* (*Aaa*) and *Aedes aegypti formosus* (*Aaf*)) from Sudan using the NADH dehydrogenase subunit 4 (*ND4*) mitochondrial gene marker. Nineteen distinct haplotypes of the *ND4* were identified in female *Aedes aegypti* mosquitoes from the study sites. The phylogenetic relationship of the 19 *ND4* haplotypes was demonstrated in a median-joining haplotype network tree with *Aaa* and *Aaf* populations found to share three haplotypes. The genetic variance (Pairwise *F_ST_* values) was estimated and found to range from 0.000 to 0.811. Isolation by distance test revealed that geographical distance was correlated to genetic variation (coefficient value (r) = 0.43). The Polar maximum likelihood tree showed the phylogenetic relationship of 91 female *Aaa* and *Aaf* from the study sites, with most of the *Aaf* haplotypes clustered in one group while most of the *Aaa* haplotypes gathered in another group, but there was an admixture of the subspecies in both clusters, especially the *Aaa* cluster. The Spatial Analysis of Molecular Variance (SAMOVA) test revealed that the eight populations clustered into two phylogeographic groups/clusters of the two subspecies populations. The 2 *Aedes aegypti* subspecies seemed not to be totally separated geographically with gene flow among the populations.

## 1. Introduction

Arboviral infections have been identified as a global health issue, including life-threatening diseases such as dengue fever, yellow fever, zika, chikungunya, and others. Dengue viruses infect an estimated 400 million and kill 10,000–20,000 people per year [1]. Sudan has recently become more concerned about arboviral diseases, and several dengue outbreaks have been recorded around the country in recent years [2,3]. 

*Aedes aegypti* has played a principal role in arboviral disease outbreaks in Sudan since the 1950s, and it is widely distributed in all arboviral disease locations in the country [4]. It is believed to exist in two forms/subspecies in Africa [5]. The wild dark form, *Aedes aegypti formosus* (*Aaf*), has only been documented from Sub-Saharan Africa [6], whereas the domestic form, *Aedes aegypti aegypti* (*Aaa*), which is lighter in color, is distributed in tropical and subtropical countries [7]. 

Our recent study revealed the presence of both forms in Sudan, with *Aaa* distributed in eastern and central Sudan east of the Nile River and *Aaf* distributed in western Sudan and west of the Nile River [8]. The study also revealed that the two subspecies seemed to be genetically distinct when using cytochrome c oxidase subunit 1 (*CO1*), and their geographical distribution might be one of the reasons for the genetic separation.

The two subspecies/forms are known to have varied vector competence in spreading arboviruses, with the domestic form (*Aaa*) having a higher vector competence in transmitting both yellow fever and dengue viruses than the wild form (*Aaf*) [9,10] The variation in vectorial capacity is one of the primary reasons for researching their genetic differences to understand disease transmission and epidemiology better. Several studies have utilized various genetic markers such as nuclear DNA, microsatellite, and mitochondrial DNA to better understand the genetic structure and variability of *Ae. aegypti* populations [11,12,13]. Mitochondrial DNA evolves at a faster rate than nuclear DNA, which makes it suitable for phylogenetic research investigations. Mitochondrial DNA has been shown to be an efficient genetic marker for determining species gene flow, and it is frequently used in population genetic studies, particularly in *Ae. aegypti* populations across the globe [7,13,14,15,16].

In a study by Ashfaq et al. [17], the mitochondrial cytochrome c oxidase subunit 1 (*CO1*) gene was found to be an effective genetic marker for genetic differentiation studies. The mitochondrial NADH dehydrogenase subunit 4 (*ND4*) gene marker in *Ae. aegypti* has also been widely used for genetic variation studies around the world, including Brazil [18,19], Peru [12] and Bolivia [20]. Paupy et al. [21] used mtDNA-*ND4* to study the genetic variation between sylvan and domestic *Ae. aegypti* populations in Cameron and discovered that the two populations are structured by limited gene flow between them.

Recently, our research group studied the subspecies diversity of *Ae. aegypti* using *CO1* mitochondrial marker [8]. However, for a comprehensive understanding of the genetic structure and composition of this disease vector, there is a need to assess the genetic diversity using other molecular markers such as *ND4*.

The two *Ae. aegypti* subspecies are found in outbreak locations in Sudan, implying that they are involved in disease outbreaks. In this work, we studied the genetic differences among the *Ae. aegypti* forms *Aaa* and *Aaf* in Sudan using *ND4* mitochondrial marker.

## 2. Methods

### 2.1. Study Area and Sample Collection

The current study was conducted in eight study sites spread across Sudan. Selection of the sites was based on the prevalence of *Ae. aegypti* and outbreaks of arboviral diseases as outlined in Abuelmaali et al. [8]. The sites were Port Sudan (main coastal port city in the Red Sea State, Tokar (Red Sea State), Kassala (Kassala State in eastern Sudan), Barakat (located in Gezira State in the middle of the country with a big agricultural scheme in the area), Kadugli city (in South Kordofan State at the northern edge of the White Nile in southern Sudan), Nyala (capital of South Darfur state in the south-west of Sudan), Al Junaynah (West Darfur state), and Al Fasher (North Darfur state) (Figure 1).

*Aedes* eggs, larvae and pupae were sampled from different breeding habitats (including clay pots, cement reservoirs, barrels, jerrycans, flowerpots, cement reservoirs, tires, and bathtubs) in the study area from 2014 to 2017. At each site, there were five sampling points (inter-point distance of at least 1 km) to avoid sibling bias in the sample collection. The samples were transferred to the insectarium for sorting and morphological identification [22] (Table 1). The samples were maintained at a temperature of 25 °C, relative humidity of 80–90% and a 12 h Light:12 h Dark photoperiod and fed with larval food until adult emergence. Identification of adults into subspecies was performed using the key of Huang [23] based on the presence or absence of a batch of median white scales on the first abdominal tergite. Each sample was preserved in a labeled microfuge tube (Thermo Fisher Scientific, MA, USA) containing 70% isopropanol and stored at −20 °C.

### 2.2. DNA Extraction, Amplification and Sequencing of the ND4 Gene

Genomic DNA (gDNA) of each female mosquito was extracted (total of ninety-two samples) using DNeasy Blood and Tissue Extraction System (Qiagen, Hilden, Germany) according to the producer’s procedures. Nanodrop Quawell UV spectrophotometer Q3000 (Quawell, San Jose, CA, USA) was used to assess the quality of the gDNA. The extracted gDNA was preserved at −20 °C prior to use in the amplification process.

The mitochondrial marker NADH dehydrogenase subunit 4 (*ND4*) was utilized to investigate genetic variations among the *Aedes aegypti* subspecies (Table 1). The forward primer (*ND4*-F) 5′-ATTGCCTAAGGCTCATGTAG-3′ and the reverse primer (*ND4*-R) 5′-TCGGCTTCCTAGTCGTTCAT-3′ were used to amplify a 400 bp partial *ND4* gene fragment as described by Da Costa-da-Silva et al. [12]. The reaction volume of 25 μL contained 5 μL of 10× Go Taq Buffer, 1.5 μL of 25 mM MgCl_2_, 0.5 μL of 25 mM dNTP, 0.3 µM of each of the primers *ND4-F* and *ND4-R*, 0.25 μL of Taq polymerase, and 1 μL (>50 ng) of sample DNA. Thermal cycling conditions were as follows: a 2-min denaturation stage at 94 °C, followed by 35 cycles at 94 °C for 1 min, 56 °C for 30 s, 72 °C for 1 min and a 10 min final extension at 72 °C [20].

The amplified products were loaded on a 1.5% agarose gel electrophoresis (Bio-Rad Laboratories, Inc., Hercules, CA, USA) stained with Red Safe dye (iNtRON, Burlington, WA, USA) for one hour and visualized using the UV light gel documentation system. The amplified products were purified using the QIAquick PCR purification kit (Qiagen, Hilden Germany) according to the manufacturer’s procedure before being sent to NHK Bioscience Solutions Sdn. Bhd (NHK Bioscience, Macrogen, Republic of Korea) for DNA Sanger sequencing on an ABI3730XL DNA Analyzer.

The sequences were aligned using ClustalW in the Molecular Evolutionary Genetics Analysis software (MEGA) v7.0 [24] and used in homology search using BLAST in the GenBank database.

DNAsp v5.10.1 software was used to identify distinct haplotypes for the *Aedes aegypti* subspecies for the *ND4* mitochondrial gene [25] and were assigned the accession numbers OM321307 to OM321325 in the GenBank database. https://www.ncbi.nlm.nih.gov/nuccore/OM321307 (accessed on 22 October 2022)

### 2.3. Genetic Variations

The haplotypes number (*h*), number of polymorphic sites (*S*), haplotype diversity (*Hd*), and nucleotide diversity (π) indices were assessed in all populations using Arlequin v3.5 [26]. The Analysis of Molecular Variance (AMOVA) was used to calculate the molecular variations ‘between groups’ and ‘within groups’ in Arlequin v3.5 [26]. The pairwise genetic distances (*F_ST_*) and their significance between the various populations were calculated. The geographical inter-site distance was determined using Google Earth TM’s ruler feature.

Computation of the Genetic isolation by distance (IBD) with the Mantel test was used to determine the inter-site correlation between genetic variations and geographical distance in Arlequin v3.5 [26]. Pairwise geographical distance was used to determine genetic isolation by distance (IBD) and coordinated independently to pairwise *F_ST_* to generate a linear regression relationship.

### 2.4. Phylogenetic Relationship between Populations

A polar unrooted maximum-likelihood phylogenetic tree was generated in IQ-TREE1.6.9 [27] and constructed with Figtree v1.4.0 [28]. Model Finder was used to determine the best model (HKY+F+I) for the polar tree [29]. NETWORK v10.00 [30] (Fluxus Technology Ltd., Hong Kong, 1999–2010, 1999) was used to analyze the association between the haplotypes.

The phylogenetic relationship of the nineteen haplotypes was further examined using Neighbor-Joining method [31] performed in MEGA11 [32]. The evolutionary history of the taxa under study is assumed to be represented by the bootstrap consensus tree inferred from 500 replicates. The Tamura 3-parameter method was used to calculate the evolutionary distances [33]. Branches corresponding to partitions that have been replicated in less than 50% of bootstrap replicates were collapsed.

### 2.5. Population Genetic Structure

SAMOVA (Spatial Analyses of Molecular Variance) v2.0 was used to understand the genetic barriers on population groups and their structure [34]. It was determined by categorizing geographically comparable populations and effectively separating the highest F_CT_ (inter-group diversity), which was maximum when the groups were two. Accordingly, we inferred that two-group clustering is the best grouping (best homogeneity among groups). Following that, the results of the SAMOVA were used for the Analyses of Molecular Variance (AMOVA) using Arlequin v3.5. Arlequin v3.5 was also used to detect population growth and deviation from neutrality using Tajima’s D and Fu’s F statistics [26].

## 3. Results

### 3.1. Identification and Haplotype Analysis of Mitochondrial ND4 Genes

A hundred samples from each site were morphologically identified as *Aedes aegypti*. Subspecies identification revealed the distribution of the samples among the 2 *Aedes aegypti* subspecies, as indicated in Table 2.

Cleaning and trimming the sequenced samples of the 400 bp *ND4* amplified gene fragment resulted in a 172 bp sequence which was then subjected to haplotype analysis.

Overall, 19 distinct haplotypes of the *ND4* mitochondrial gene were detected in this study. The haplotype number and their diversity are described in Table 2.

The number of haplotypes varied from eight in Tokar, five in Kadugli and Kassala, four in Nyala and Al Junaynah, with the least being two in Port Sudan and Barakat, and one in Al Fasher (Table 2).

### 3.2. Genetic Variability in ND4 Mitochondrial Genes of Aedes aegypti

The *Aaa* subspecies had a higher number of polymorphic sites as well as nucleotide diversity compared to *Aaf* (28 and 20 polymorphic sites in *Aaa* and *Aaf,* respectively). The haplotype diversity was different and inconsistent within *Aaa* and *Aaf* populations, the highest in the Tokar site (0.956), followed by Nyala (0.75) and the lowest also among *Aaf* populations (0.00 in Al Fasher) which had only one haplotype in its population followed by Barakat *Aaa* population (0.2). In *Aaa* populations, the haplotype diversity was moderate to high in Kassala (0.539) and relatively low in Port Sudan (0.286) (Table 2).

### 3.3. Phylogenetic Relationships in Aedes aegypti Subspecies

The phylogenetic relationship of the 19 *ND4* haplotypes was demonstrated in a median-joining haplotype network tree (Figure 2). Eleven haplotypes were site-specific haplotypes.

Hap 1 showed higher frequency and included mosquitoes from all *Aaf* populations and the Kassala population of *Aaa*, followed by Hap 5, which was found in the and Kassala *Aaa* populations. Ten haplotypes appeared only in a single mosquito, while many mosquitoes shared nine haplotypes. Hap 1 was found in 73.6% of *Aaf* populations, while Hap 5 was observed in 29.3% of *Aaa* populations and appeared in only two populations (Kassala and Barakat), in addition to Hap 11, which appeared in Port Sudan, Tokar and Kassala and was found in 19.5% of all the *Aaa* samples.

The *Aaa* and *Aaf* populations shared three haplotypes (Figure 2). Hap 1 was detected in 39 *Aaf* samples from Nyala, Al Fashir, Kadugli and Al Junaynah and appeared in 7 *Aaa* samples in Kassala. Hap 4 was found in three *Aaf* samples from Nyala and Al Junaynah and two *Aaa* samples in Barakat. Hap 13 was observed in one *Aaf* sample in Kadugli and two *Aaa* samples in Kassala.

### 3.4. Pairwise F_ST_ Genetic Variations and Isolation by Distance Results:

The *ND4* genetic variance of the subpopulation (Pairwise *F_ST_* values) ranged from 0.000 to 0.811 (Table 3). Pairwise *F_ST_* values between all populations showed mostly significance (*p* < 0.05), as shown in Table 3. Populations of Tokar (*Aaa*) and Nyala (*Aaf*) had no genetic differences between them, with a pairwise *F_ST_* value of 0.000 (Table 3). In contrast, the Port Sudan *Aaa* population revealed a high genetic difference from the Barakat *Aaa* population (0.74).

The Mantel test revealed a significant (*p* = 0.02) relationship between genetic variation and geographical distance (correlation coefficient = 0.43) (Figure 3). Conversely, a few sites varied from this correlation, for example, the lowest pairwise *F_ST_* values between Kadugli and Nyala, and Nyala and Tokar with long distances between the sites with the values (*F_ST_* = 0.000, distance = 556 km) and (*F_ST_* = 0.000, distance = 1550 km), respectively. Some populations also had high *F_ST_* values despite being geographically closer, such as Port Sudan and Barakat (*F_ST_* = 0.74, distance = 700 km) and Port Sudan and Kassala (*F_ST_* = 0.532, distance = 489 km) (Figure 3).

### 3.5. Population Genetics

The phylogenetic tree of *Aedes aegypti ND4* from the study sites clustered in two groups with a mixture of the two subspecies populations in the two groups (Figure 4); *Aedes aegypti* subspecies seemed not to be totally separated geographically, and the phylogenetic tree showed gene flow (mixing between the two among their populations). The neighbor-joining phylogenetic tree was bifurcated to yield two clades with an admixture of the subspecies haplotypes (Figure 5).

The *Aaf* haplotypes clustered in one group, while most of the *Aaa* haplotypes gathered in another group. Some Tokar and Kassala *Aaa* individuals who shared the same haplotypes as the *Aaf* populations clustered together. On the other hand, a few samples of *Aaf* from Kadugli, Nyala and Al Junaynah clustered with *Aaa* populations (Figure 4).

The genetic structure and the best-fit grouping pattern were estimated in SAMOVA, which indicated that the eight populations are divided into two phylogeographic groups. The first group includes *Aaf* populations from Nyala, Kadugli, Al Fasher and Al Junaynah, while the second comprises *Aaa* populations from Port Sudan, Kassala, Barakat and Tokar.

The genetic variance (AMOVA test) for the *ND4* mitochondrial gene of the *Ae. aegypti* populations showed 26.64% and 48.83% genetic variance among and within populations of the two groups (Table 4).

### 3.6. Neutrality Test and Natural Selection

Tokar, Kassala, Nyala and Kadugli had positive Tajima’s D values indicating a natural balancing selection or population substructuring, while Port Sudan, Barakat and Al Junaynah had negative values indicating a recent directional selection or recent population growth (Table 2). Positive Fu’s *F_S_* results were recorded in Port Sudan, Kassala, Barakat, Kadugli and Nyala, indicating a recent population bottleneck or over-dominant selection. The Al Fasher site had no value because there was only one haplotype present, whereas Tokar and Al Junaynah had negative values, indicating a recent population expansion or genetic hitchhiking. (Table 2).

## 4. Discussion

*Aedes aegypti* is widely distributed in Sudan and is regarded as the primary vector for a variety of arboviral diseases such as yellow fever, dengue fever, and chikungunya [2,35,36,37].

Considering the significance of investigating the genetic diversity among *Ae. aegypti* vector populations and its contribution to the knowledge of the dissemination and transmission of arbovirus illnesses and epidemics in Sudan, this study reports the genetic variations of *Ae. aegypti* in Sudan using *ND4* mitochondrial gene. Since the identification of the *Ae. aegypti* subspecies/forms [5], studies on their role in arboviruses transmission in endemic areas in Africa and Sudan are limited. Although the vector competency of *Aaf* is lower than *Aaa*, various elements can influence the vectorial capability, including the distribution, density and life span of the vector, its adaptability to the environment and the human-biting index [9,10,38].

The findings of this study, using the *ND4* mitochondrial gene, indicated gene flow among the populations of the *Aedes aegypti* subspecies, suggesting that they are not entirely genetically isolated. The genetic variations between the different populations revealed high prevalence in most populations; however, some of the *Aaf* populations have no genetic difference from other *Aaa* populations (such as Tokar and Nyala, *F_ST_* value = 0.000). On the other hand, variations between some *Aaa* populations were very high (such as Port Sudan and Barakat, *F_ST_* = 0.740). This finding contrasts our previous study, which indicated high genetic variation between most of the subspecies’ populations, although, in both studies, the Tokar and Barakat *Aaa* populations showed the least variations with *Aaf* populations. This might be attributed to their geographical distance (closer) to the western parts of the country where the *Aaf* populations are distributed. Moreover, the high *F_ST_* values between many *Aaf* and *Aaa* populations, such as at the Port Sudan and Al Fasher sites (*F_ST_* = 0.811), may be attributed to their geographical isolation (1424 km between them). The significant correlation between the geographical distance and the genetic variation revealed in isolation by distance tests in both studies is probably one of the main reasons for the high genetic variations between populations.

Generally, *Ae. aegypti* subspecies seemed not to be totally separated (structured) in the phylogenetic tree, and the results showed gene flow among their populations. The two subspecies are clustered into two groups on the polar phylogenetic tree, with *Aaa* samples from Kassala mixing with the *Aaf* populations cluster and many samples of *Aaf* populations clustered with the *Aaa* cluster, as shown in Figure 4. At the haplotype level, admixing of the subspecies in the neighbor-joining tree (Figure 5) aligns with the fact that there is a gene flow among the populations.

SAMOVA analysis revealed that the percentage of variation between the two subspecies groups was only 26.64% which is lower than the percentage variation (39.2%) in our last study using the *CO1* mitochondrial gene [8]. This could be due to the mixing of the two subspecies through human movement and transportation, disproving their isolation and genetic separation.

In accordance with Abuelmaali et al. [8], the population grouping showed that the populations congregated phylogeographically into two groups of subspecies. It is difficult to tell whether this grouping reflects a true difference between the two subspecies or was caused by limited gene flow in some populations due to geographic distances. Microsatellite markers used in a study by Gloria-Soria et al. [39] that compared sympatric and allopatric *Ae. aegypti* populations found that the two subspecies are genetically distinct. Hence the findings of our study indicate that there is probably a recent gene flow between the two subspecies as *Aaf* encroached into human territories where *Aaa* already exists. Powell et al. [40] determined, based on genetic data, that the two subspecies were completely isolated and had no gene flow around 400–550 years ago.

It is worth mentioning that *Aaf* showed relatively lower genetic variations compared to *Aaa* populations, which agreed with the findings of Brown et al. [11] and Powell and Tabachnick [6]. Some ecological conditions could be responsible for this since our *Aaf* population collection sites were generally drier than these authors’ sites. The severe aridity, when combined with the huge physical distances between our study locations, inhibits migration and gene exchange between the populations.

In this study, the genetic variation in the *Ae. aegypti* populations varied by location, as AMOVA revealed a percentage of 48.83% for population variations. The *Aaa* population in the coastal city of Port Sudan had the lowest variation in the *CO1* gene, with five haplotypes reported at the site [8]. Elnour et al. [14] reported similar findings in Port Sudan city using the *CO1* mtDNA gene. The persistent high humidity by the Red Sea all year round lessens the effect of aridity on the mosquito populations and could explain the decreased genetic variation at Port Sudan. Despite belonging to the same subspecies, Port Sudan (*Aaa*) and Barakat (*Aaa*) displayed genetic differentiation from one another, according to examinations of the genetic differentiation (*F_ST_* values) between sites. Additionally, there were no genetic differences between the Tokar *Aaa* population and some *Aaf* populations. It may be possible to split individuals/populations within a group into a smaller group or subpopulation in light of the high levels of diversity among individuals and populations within groupings.

The haplotype numbers from this study were different from our previous study using *CO1* [8]. Al Fasher mosquito population had the lowest diversity, with only one haplotype recorded at the site. The difference in the haplotype number might be attributed to the difference in the number of sequenced samples in the study as well as the different genes and sequences length.

The Tajima D and Fu’s F statistics predicted a stronger natural selection impact at some sites across the country, such as the Tokar site, which showed a positive Tajima value.

Finally, this study reveals uncertainties about the distinctiveness of the genetic structure of the two subspecies, contrary to our previous study using the *CO1* mitochondrial gene [8], which revealed that they were distinct and genetically structured. This present study, however, has some limitations, such as the uneven sample size in the sites, which may cause some bias in the results.

In Sudan and other African Sahelian countries, more research is needed to decipher the role of the subspecies in the spread of *Aedes aegypti*-borne diseases. An understanding of the ecological factors that influence the distribution and abundance of the subspecies, as well as control measures for the viruses transmitted by them, are also worth investigating.

## Figures and Tables

**Figure 1 insects-13-01144-f001:**
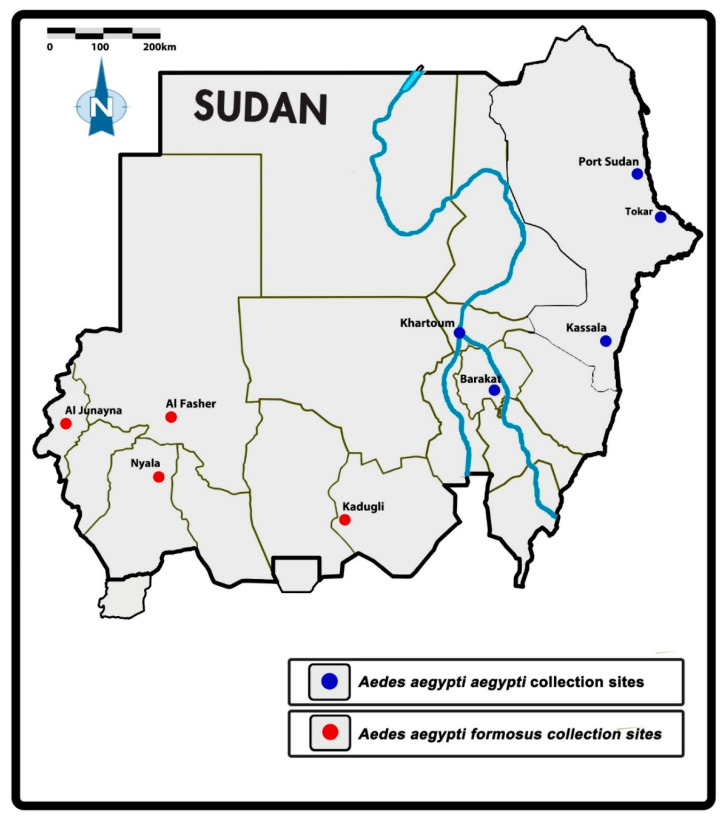
Map of the study locations in Sudan [8].

**Figure 2 insects-13-01144-f002:**
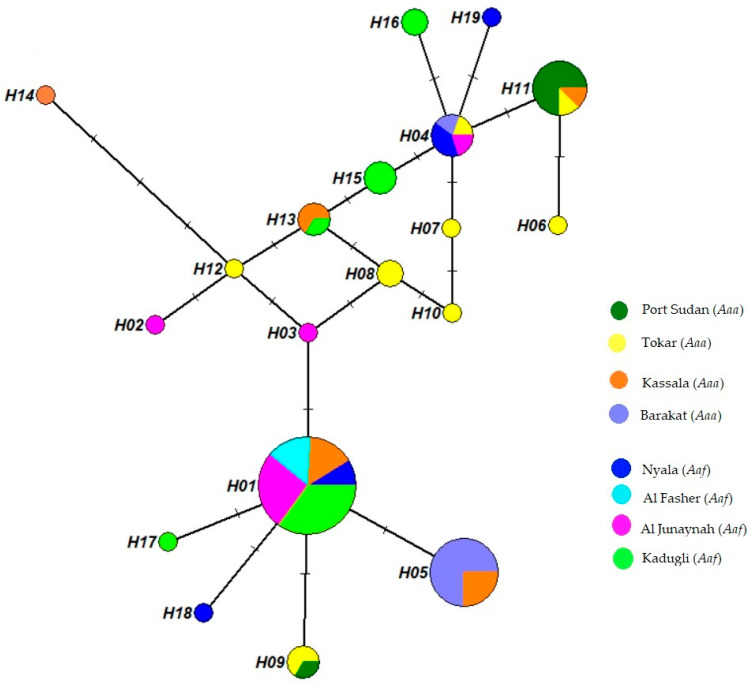
*Aedes aegypti* haplotype network tree with median-joining haplotypes. The tree was built using 19 haplotype *ND4* sequences from 8 study sites. Each circle’s size represents the frequency of the haplotype’s occurrence. A minimum of 1 mutation exists between 2 haplotypes.

**Figure 3 insects-13-01144-f003:**
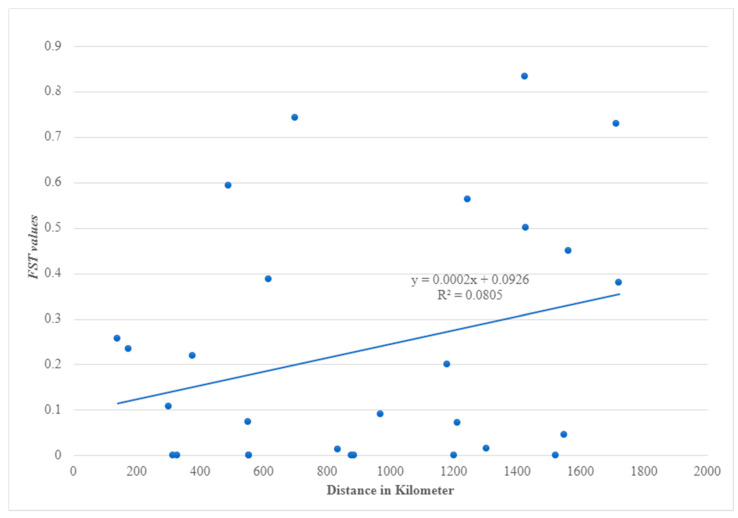
Correlation between genetic (*F_ST_*) of NADH dehydrogenase subunit 4 (*ND4*) mitochondrial gene and geographical distance (km) of *Aedes aegypti* samples from Sudan.

**Figure 4 insects-13-01144-f004:**
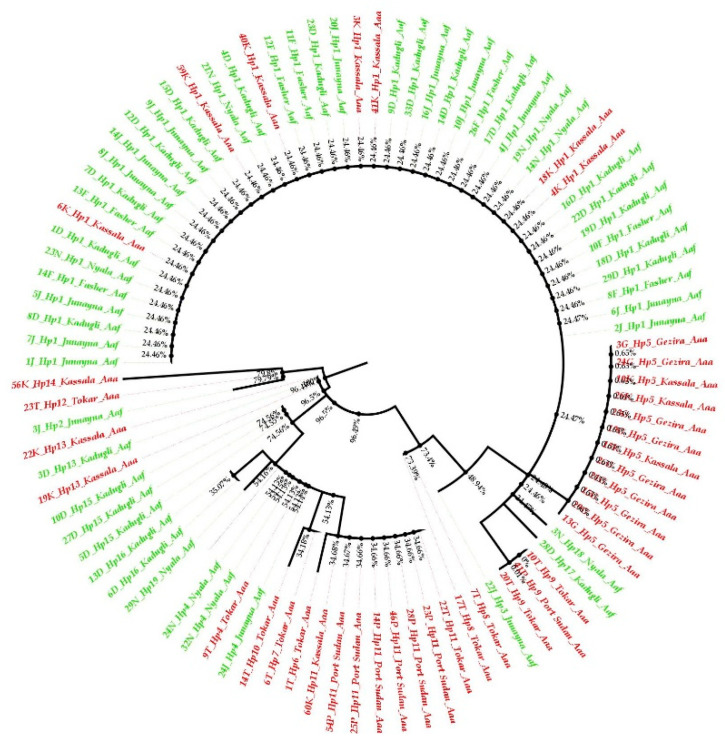
Polar maximum likelihood phylogenetic tree from 91 *ND4* mitochondrial gene sequences from eight *Aedes aegypti* subspecies populations in Sudan. *Aaa* (red color), while *Aaf* (green color).

**Figure 5 insects-13-01144-f005:**
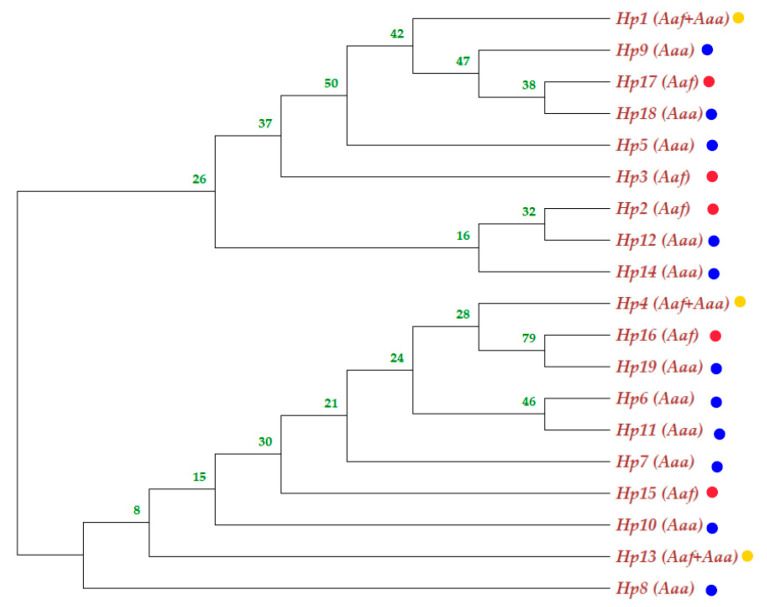
Neighbor-Joining phylogenetic tree involving nineteen haplotypes of *ND4* mitochondrial gene sequences from eight *Aedes aegypti* subspecies populations in Sudan. The yellow dot haplotypes were shared between the subspecies; the blue dot indicates *Aaa* haplotype, while the red dot indicates *Aaf* haplotype.

**Table 1 insects-13-01144-t001:** *Aedes aegypti* populations and their coordinates.

Population/Code	Year	State	Form	NO	Coordinates
Port Sudan (P)	2014	Red Sea	*Aaa*	7	19.617°37′0″ N, 37.217°13′0″ E
Tokar (T)	2016	Red Sea	*Aaa*	8	18.425°25′31″ N, 37.729°43′45″ E
Kassala (K)	2014	Kassala	*Aaa*	14	15.45°27′0″ N, 36.4°24′0″ E
Barakat (G)	2014	Gezira	*Aaa*	10	14.314°18′50.84″ N, 33.534°32′3.74″ E
Kadugli (D)	2015	South Kordofan	*Aaf*	23	11.017°1′0″ N, 29.717°43′0″ E
Nyala (N)	2016–2017	South Darfur	*Aaf*	8	12.036°2′11″ N, 24.878°52′37″ E
Al Fasher (F)	2017	North Darfur	*Aaf*	7	13.631°37′50″ N, 25.35°21′0″ E
Al Junaynah (J)	2014	West Darfur	*Aaf*	15	13.45°27′ 0″ N, 22.45°27′0″ E

NO, number of samples.

**Table 2 insects-13-01144-t002:** The haplotype number, diversity, nucleotide diversity and neutrality tests for *ND4* gene samples of *Aedes aegypti* from Sudan.

Site (Subspecies)	*N*	*S*	*H*	*Hd*	*Π*	Tajima’s D	*F_S_*
Port Sudan (*Aaa*)	7	7	2	0.286	0.012	−1.55311	3.29584
Tokar (*Aaa*)	10	7	8	0.956	0.019	1.28010	−3.31369
Kassala (*Aaa*)	14	8	5	0.539	0.014	0.29050	0.36807
Barakat (*Aaa*)	10	6	2	0.200	0.007	−1.79631	2.60670
Kadugli (*Aaf*)	23	7	5	0.510	0.012	0.14600	0.89004
Nyala (*Aaf*)	8	7	4	0.75	0.019	0.84405	1.12851
Al Fasher (*Aaf*)	7	0	1	0.000	0.000	0.000	0.00000
Al Junaynah (*Aaf*)	15	6	4	0.371	0.007	−1.36873	−0.07744

*N*, number of samples; *S*, polymorphic sites’ number, *H*, haplotype number; *Hd*, haplotype diversity; π, diversity of nucleotides; D, Tajima’s statistics; *F_S_,* Fu’s F statistics.

**Table 3 insects-13-01144-t003:** *Aedes aegypti* population divergence between *ND4* mitochondrial gene sequences (*F_ST_* value) in the different study sites from Sudan.

Site	Port Sudan	Tokar	Kassala	Barakat	Nyala	Al Fasher	Al Junaynah
Port Sudan							
Tokar	0.210 *						
Kassala	0.532 *	0.406 *					
Barakat	0.740 *	0.442 *	0.521 *				
Nyala	0.378	0.000	0.440 *	0.323 *			
Al Fasher	0.811 *	0.423 *	0.650 *	0.518 *	0.228		
Al Junaynah	0.713 *	0.325 *	0.592 *	0.388 *	0.116	0.009	
Kadugli	0.572 *	0.190 *	0.515 *	0.309 *	0.000	0.073	0.002

Port Sudan, Kassala, Barakat and Tokar sites are *Aaa* populations, while Nyala, Al Fasher, Al Junaynah and Kadugli sites are *Aaf*. The values with * have a significant *p* value.

**Table 4 insects-13-01144-t004:** Analysis of molecular variance (AMOVA) of *ND4* mitochondrial gene of *Aedes aegypti* from Sudan.

Source of Variation	df	Sum of Squares	Variance Components	Variation Percentage
Among Groups	1	7.433	0.12794	26.64
Among Individuals within Groups	6	8.694	0.11785	24.54
Within Populations	81	18.997	0.23453	48.83
Total	88	35.124	0.48031	

df, degree of freedom.

## Data Availability

Data is contained within the article.

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
