# Peer review of "Genetic Polymorphism and Phylogenetics of Aedes aegypti from Sudan Based on ND4 Mitochondrial Gene Variations"

_insects, 2022, doi:10.3390/insects13121144_

Round 1

Reviewer 1 Report

General comments

The manuscript entitled "Genetic polymorphism and phylogenetics of Aedes aegypti from Sahelian belt in Sudan based on ND4 mitochondrial gene variations" by Abuelmaali and colleagues reports the genetic differences between the two forms/subspecies of Aedes aegypti from Sahelian belt in Sudan the Aedes aegypti aegypti (Aaa) and Aedes aegypti formosus (Aaf) using the ND4 mitochondrial gene marker. The authors focus on measuring the differences between the Aaa and Aaf populations. Using the phylogeny, they could differentiate 19 ND4 haplotype clusters, being only three shared by both Aaa and Aaf populations. The genetic variance (FST) between Aaa and Aaf populations was variable, and the isolation by distance tests is strongly correlated (r = 0.7) with the genetic variance. They have strong evidence of the existence of two subspecies of Aedes aegypti from the Sahelian belt in Sudan based on the population-specific clustering of the Aaa and Aaf haplotypes and the AMOVA test.

I have no significant comments and a few minor issues for the authors to consider:

Minor Comments:

While the overall quality of the language is fine, there are some mistakes the authors should fix. Some (but not all) of which I have listed below:

Line 3, in the title, I would suggest a minor modification to “the Sahelian belt in Sudan”  instead of “Sahelian belt in Sudan.”

Lines 25, 26, 28, and 37 Please italicize the species names.

Lines 80 and 83 “mitochondrial dehydrogenase subunit-4 (ND4) and NADH dehydrogenase subunit 4 (ND4) mitochondrial

Line 89-92 I would suggest rewriting the paragraph since this is the aim of the work “However, recently our research group studied the distribution and the genetic variations of the two subspecies of Aedes aegypti in the Sudan Sahelian region using CO1 mitochondrial marker. Therefore, the need to observe the genetic diversity using other molecular markers such as ND4 to observe the similarities and explore the differences, if any.”

Line 120 I would suggest putting in supplemental material table 1 Aedes aegypti populations and their coordinates in the article's main text.

Line 126. Please indicate the total number of females from which the DNA was extracted and used in this study.

Line 234-235 Please clarify this sentence “Thirteen haplotypes were detected in a single mosquito, while six haplotypes were shared by many mosquitoes.”

Line 253 -254. In the paragraph was written, “Although some populations had high FST values despite being geographically close, such as Port 254 Sudan and Barakat (FST = 0.74, distance = 700 km) and Port Sudan and Kassala (FST = 0.59, 255 distance = 489 km).” Please consider that 700 km and 489 km is not a short distance. I suggest modifying with “being geographically closer.”

Author Response

Thank you so much for reviewing our article.

Response to reviewer 1:

While the overall quality of the language is fine, there are some mistakes the authors should fix. Some (but not all) of which I have listed below:

Thank you so much, the mistakes have been fixed as required, below are my response to each comment.

Line 3, in the title, I would suggest a minor modification to “the Sahelian belt in Sudan” instead of “Sahelian belt in Sudan.”

The title modification has been done.

Lines 25, 26, 28, and 37 Please italicize the species names.

The species names have been italicized.

Lines 80 and 83 “mitochondrial dehydrogenase subunit-4 (ND4) and NADH dehydrogenase subunit 4 (ND4) mitochondrial

It has been corrected.

Line 89-92 I would suggest rewriting the paragraph since this is the aim of the work “However, recently our research group studied the distribution and the genetic variations of the two subspecies of Aedes aegypti in the Sudan Sahelian region using CO1 mitochondrial marker. Therefore, the need to observe the genetic diversity using other molecular markers such as ND4 to observe the similarities and explore the differences, if any.”

The paragraph has been revised.

Line 120 I would suggest putting in supplemental material table 1 Aedes aegypti populations and their coordinates in the article's main text.

Thank you for your comment, however, I think table one showed essential data to understand the overall manuscript, so I kept it in the main text.

Line 126. Please indicate the total number of females from which the DNA was extracted and used in this study.

The total number has been added.

Line 234-235 Please clarify this sentence “Thirteen haplotypes were detected in a single mosquito, while six haplotypes were shared by many mosquitoes.”

The sentence has been clarified.

Line 253 -254. In the paragraph was written, “Although some populations had high FST values despite being geographically close, such as Port 254 Sudan and Barakat (FST = 0.74, distance = 700 km) and Port Sudan and Kassala (FST = 0.59, 255 distance = 489 km).” Please consider that 700 km and 489 km is not a short distance. I suggest modifying with “being geographically closer.”

The sentence has been modified

Reviewer 2 Report

Major comments:

·         A similar study used different mitochondrial gene (COI) has been conducted in Sudan, therefore, the current study should indicate what are new findings and differences. I did not see much comparisons between two studies to show the importance and new findings of the current study. Therefore, authors need to explain why they should use ND4 gene (still mitochondrial gene)?

·         The mosquito sampling plan is not well described. Collection methods did not mention how to prevent sibling bias since Aedes mosquitoes lay their eggs in several nearby containers. This is very important because it could result a homogenous population.

·         The number of analyzed individuals is comparable between sites, some are very low (7-10 individuals), it could make unreliable in the genetic indices.

·         The information in Table 2 need to be checked carefully including haplotype diversity (e.g. 8 individuals, 4 haplotypes but Hd is 1, I think something went wrong here), two indications for S, text formatting, …

·         The analyses of genetic differentiation between sites showed that site P (Aaa) and T(Aaa) genetically differentiated from one another even K(Aaa) and G(Aaa). Further, K&G has no significantly genetic difference with other Aaf sites. Even single site P or T also significant different to one another. It could explain why the FSC (among individuals/populations within groups) is very high, that indicate for individuals/populations within a groups can be separated in a smaller group. Therefore, further interpretations/analyses of population structure are required.

·         The phylogenetic tree should go with bootstrap numbers in each branch to indicate the possibility and accuracy, otherwise we cannot conclude the genetic structure between two forms.

·         Even without bootstrap number, we can see the genetic structure between two morphological forms is not clear since they’re mixing in the tree. Only the frequency of them in each geographical region is different. Therefore, the conclusion in this study need to be adjusted.

Minor comments:

·         Line 44: Zika viruses are not disease, change to Zika

·         Line 234, 10 haplotypes can be seen in hap network for single individual

·         Based on the Figure 2, the Haplotype 5 has been found only in Aaa individuals, so the information in line 233 need to be checked.

·         All the words in bold in table need to be explained for why they’re in bold, if there is no indication, they should be changed to normal. Format the font in the Table 1

·         Species, genes and some genetic indices are in Italic form

·         Figure 1 should be changed by a better resolution one

·         Sentence “A total of 19 distinct haplotypes of the ND4 mitochondrial gene were detected in all 203 female Aedes aegypti mosquitoes from all the study sites.” Is not clear, please rewrite.

Author Response

Thank you so much for taking the time to review our manuscript.

Response to reviewer 2

  • A similar study used different mitochondrial gene (COI) has been conducted in Sudan, therefore, the current study should indicate what are new findings and differences. I did not see much comparisons between two studies to show the importance and new findings of the current study. Therefore, authors need to explain why they should use ND4 gene (still mitochondrial gene)?

                  The manuscript has been modified and more comparisons with the last study have been added, all changes were marked by track changes.

  • The mosquito sampling plan is not well described.Collection methods did not mention how to prevent sibling bias since Aedes mosquitoes lay their eggs in several nearby containers. This is very important because it could result a homogenous population.

                 The mosquito collection plan was added to the methodology part.

  • The number of analyzed individuals is comparable between sites, some are very low (7-10 individuals), it could make unreliable in the genetic indices.

We acknowledge that this is possible, and it is part of the study limitations, however, similar publications/studies from PubMed have low numbers on some sites (some of them are cited below).

Ali KHA, El-Badry AA, Ali MA, El-Sayed WSM, El-Beshbishy HA. Phylogenetic Analysis of Aedes aegypti Based on Mitochondrial ND4 Gene Sequences in Almadinah, Saudi Arabia. Iran J Biotechnol. 2016 Jun;14(2):58-62. doi: 10.15171/ijb.1329. PMID: 28959327; PMCID: PMC5435033.

Rašić G, Schama R, Powell R, Maciel-de Freitas R, Endersby-Harshman NM, Filipović I, Sylvestre G, Máspero RC, Hoffmann AA. Contrasting genetic structure between mitochondrial and nuclear markers in the dengue fever mosquito from Rio de Janeiro: implications for vector control. Evol Appl. 2015 Oct;8(9):901-15. doi: 10.1111/eva.12301. Epub 2015 Sep 7. PMID: 26495042; PMCID: PMC4610386.

  • The information in Table 2 needs to be checked carefully including haplotype diversity (e.g. 8 individuals, 4 haplotypes but Hd is 1, I think something went wrong here), two indications for S, text formatting, …

          The information in table 2 has been revised and corrected.

  • The analyses of genetic differentiation between sites showed that site P (Aaa) and T(Aaa) genetically differentiated from one another even K(Aaa) and G(Aaa). Further, K&G has no significantly genetic difference with other Aaf sites. Even single site P or T also significant different to one another. It could explain why the FSC(among individuals/populations within groups) is very high, that indicate for individuals/populations within a groups can be separated in a smaller group. Therefore, further interpretations/analyses of population structure are required.

         Further interpretation has been added (line 411 to 418)

  • The phylogenetic tree should go with bootstrap numbers in each branch to indicate the possibility and accuracy, otherwise we cannot conclude the genetic structure between two forms.

         The phylogenetic tree has been modified as required.

  • Even without bootstrap number, we can see the genetic structure between two morphological forms is not clear since they’re mixing in the tree. Only the frequency of them in each geographical region is different. Therefore, the conclusion in this study need to be adjusted.

          The conclusion has been adjusted.

Minor comments:

  • Line 44: Zika viruses are not disease, change to Zika

         It has been changed.

  • Line 234, 10 haplotypes can be seen in hap network for single individual

          The sentence has been clarified.

  • Based on the Figure 2, the Haplotype 5 has been found only in Aaa individuals, so the information in line 233 need to be checked.

         The information has been corrected.

  • All the words in bold in table need to be explained for why they’re in bold, if there is no indication, they should be changed to normal. Format the font in the Table 1

         The tables have been formatted as required.

  • Species, genes, and some genetic indices are in the Italic form

         The has been modified in italic.

  • Figure 1 should be changed by a better resolution one

         Figure 1 has been changed to a better one.

  • Sentence “A total of 19 distinct haplotypes of the ND4 mitochondrial gene were detected in all 203 female Aedes aegypti mosquitoes from all the study sites.” Is not clear, please rewrite.

         The sentence has been rewritten.

Round 2

Reviewer 2 Report

1.       Because the sampling size is imbalanced between sites and low in number that can cause inaccurate results, for example, the number of haplotypes, diversity indices, demography and population structure. Therefore, I think the number of individuals on several sites should be increased, to at least 10.

"They mentioned several studies using a low number of individuals, the first one I think not a high-ranking journal, and the second one used multiple genes so they could reduce the bias from the low number of individuals."

2.       They should use only haplotypes to infer the phylogenetic tree rather than all individual sequences to show better resolution of the phylogenetic tree. The current tree is similar to the haplotype network. If they do that, I think the genetic structure of Ae. aegypti can be clearly seen, maybe three or more clades. Further, why do they need to show two different trees, green and blue?

3.       Since genetic variation is not significant, the high exchange of genes and its consequences should be discussed.

4.       The comparison between the current study and with previous one using COI does not reflect the new critical insights into science.

5. Check every sentence for grammar and meaning, many are redundant and hard to understand. (e.g. line 35-36)

Lines 43-44: “The genetic variance (Pairwise FST values) was estimated and 43 found to range from 0.000 to 0.811.” but in Table 2, the highest Hd is 0.956, check data in sentence line 247 as well.

Lines 188-192: should move to the results part for the two geographical groups.

Lines 313-314: the two subspecies are not clustered in two clades, they are mixed.

Author Response

Response to reviewer:

  1. Because the sampling size is imbalanced between sites and low in number that can cause inaccurate results, for example, the number of haplotypes, diversity indices, demography and population structure. Therefore, I think the number of individuals on several sites should be increased, to at least 10.

"They mentioned several studies using a low number of individuals, the first one I think not a high-ranking journal, and the second one used multiple genes so they could reduce the bias from the low number of individuals."

We acknowledge your comment as a limitation of our study and have included this limitation statement in the manuscript (lines….). However, a few studies (published in high impacted factors journals) also used similar numbers (less than 10 samples in some sites). Below are two examples:

  1. Elnour, M. A. B., Moustafa, M. A. M., Khogali, R., Azrag, R. S., Alanazi, A. D., Kheir, A., ... & Salim, B. (2020). Distinct haplotypes and free movement of Aedes aegypti in Port Sudan, Sudan. Journal of Applied Entomology144(9), 817-823.
  2. Escobar, D., Ortiz, B., Urrutia, O., & Fontecha, G. (2022). Genetic Diversity among Four Populations of Aedes aegypti(Diptera: Culicidae) from Honduras as Revealed by Mitochondrial DNA Cytochrome Oxidase
  3. Pathogens (Basel, Switzerland), 11(6), 620. https://doi.org/10.3390/pathogens11060620Ponce, P., Muñoz-Tobar, S., Carrazco-Montalvo, A., Villota, S. D., Coloma, J., Wang, C., Holechek, S., & Cevallos, V. (2021). Two Haplotypes of Aedes aegyptiDetected by ND4 Mitochondrial Marker in Three Regions of Ecuador. Insects, 12(3), 200. https://doi.org/10.3390/insects12030200

  1. They should use only haplotypes to infer the phylogenetic tree rather than all individual sequences to show better resolution of the phylogenetic tree. The current tree is similar to the haplotype network. If they do that, I think the genetic structure of Ae. aegypti can be clearly seen, maybe three or more clades. Further, why do they need to show two different trees, green and blue?

The phylogenetic tree of haplotypes was drawn using MEGA software and shown in figure 5

  1. Since genetic variation is not significant, the high exchange of genes and its consequences should be discussed.

The genetic variation is not significant between the subspecies (there is no genetic structure between them). More discussion was added (Page 13 &14)

  1. The comparison between the current study and with previous one using COI does not reflect the new critical insights into science.

More discussion was added to clarify the new finding of this study compared to the CO1 study (Lines 1548-1560).

  1. Check every sentence for grammar and meaning, many are redundant and hard to understand. (e.g. line 35-36)

The sentence has been checked and revised.

Lines 43-44: “The genetic variance (Pairwise FST values) was estimated and 43 found to range from 0.000 to 0.811.” but in Table 2, the highest Hd is 0.956, check data in sentence line 247 as well.

The sentence has been revised and corrected as indicated.

Lines 188-192: should move to the results part for the two geographical groups.

It has been added to results.

Lines 313-314: the two subspecies are not clustered in two clades, they are mixed.

The sentence has been checked and modified.
